

# Identification of a six-gene metabolic signature predicting overall survival for patients with lung adenocarcinoma

Yubo Cao[1], Xiaomei Lu[2], Yue Li[1], Jia Fu[1], Hongyuan Li[1], Xiulin Li[1], Ziyou Chang[1] and Sa Liu[1]

[1] Department of Medical Oncology, The Fourth Affiliated Hospital of China Medical University, Shenyang, China
[2] Department of Pathophysiology, China Medical University, Shenyang, China

Corresponding authors
Yubo Cao, ybcao@cmu.edu.cn,
cybsweeter@126.com
Sa Liu, liusa118@163.com

## ABSTRACT

**Background.** Lung cancer is the leading cause of cancer-related deaths worldwide. Lung adenocarcinoma (LUAD) is one of the main subtypes of lung cancer. Hundreds of metabolic genes are altered consistently in LUAD; however, their prognostic role remains to be explored. This study aimed to establish a molecular signature that can predict the prognosis in patients with LUAD based on metabolic gene expression.
**Methods.** The transcriptome expression profiles and corresponding clinical information of LUAD were obtained from The Cancer Genome Atlas and Gene Expression Omnibus databases. The differentially expressed genes (DEGs) between LUAD and paired non-tumor samples were identified by the Wilcoxon rank sum test. Univariate Cox regression analysis and the lasso Cox regression model were used to construct the best-prognosis molecular signature. A nomogram was established comprising the prognostic model for predicting overall survival. To validate the prognostic ability of the molecular signature and the nomogram, the Kaplan–Meier survival analysis, Cox proportional hazards model, and receiver operating characteristic analysis were used.
**Results.** The six-gene molecular signature (*PFKP, PKM, TPI1, LDHA, PTGES, and TYMS*) from the DEGs was constructed to predict the prognosis. The molecular signature demonstrated a robust independent prognostic ability in the training and validation sets. The nomogram including the prognostic model had a greater predictive accuracy than previous systems. Furthermore, a gene set enrichment analysis revealed several significantly enriched metabolic pathways, which suggests a correlation of the molecular signature with metabolic systems and may help explain the underlying mechanisms.
**Conclusions.** Our study identified a novel six-gene metabolic signature for LUAD prognosis prediction. The molecular signature could reflect the dysregulated metabolic microenvironment, provide potential biomarkers for predicting prognosis, and indicate potential novel metabolic molecular-targeted therapies.

## INTRODUCTION

Lung cancer is the leading cause of cancer-related deaths worldwide, accounting for nearly 20% of all cancer deaths (*Bray et al., 2018*). Lung adenocarcinoma (LUAD) is one of the main subtypes of lung cancer (*Travis, 2020*), accounting for more than 40% of lung cancer cases (*Hutchinson et al., 2019*), and its relative frequency is increasing (*Twardella et al., 2018*). Despite great improvements in the treatment of LUAD, the prognosis in patients with LUAD remains poor owing to the lack of early detection and effective individual therapies (*Dolly et al., 2017*). Therefore, exploring prognostic biomarkers is a critical need to help predict prognosis in LUAD and to design individual therapies. Until now, most prognostic models were based on clinical characteristics (e.g., age, sex, TNM stage, vascular tumor invasion, and organization classification) or a single molecular biomarker, such as carcinoembryonic antigen and epidermal growth factor receptor. However, these prognostic models have limited power for predicting prognosis because of the complicated molecular mechanisms of LUAD development and progression. Therefore, it is important to explore the mechanism of LUAD pathology in more depth using bioinformatics to construct prognostic models that predict the patients' prognosis more accurately.

Metabolic reprogramming is one of the hallmarks of cancer (*Faubert, Solmonson & DeBerardinis, 2020*), which takes place from the onset and throughout the development of cancer (*Chang, Fang & Gu, 2020*). It plays an important role in the progression, metastasis, depressed immunity, and therapy resistance of cancer (*Lane, Higashi & Fan, 2019*). Metabolic reprogramming has been widely accepted as the basis for the discovery of novel tumor biomarkers. *Satriano et al. (2019)* observed that metabolic rearrangement played an important role in predicting the prognosis in patients with primary liver cancers. *Chen et al. (2019)* revealed that reprogrammed tumor glucose metabolism could promote cancer stemness and result in poor prognosis in breast cancer patients. There are hundreds of metabolic genes that consistently have an altered expression in LUAD (*Asavasupreechar et al., 2019*; *Vanhove et al., 2019*); however, their roles and mechanisms of action remain unclear. This study investigated the role of abnormal metabolism in predicting the prognosis in patients with LUAD.

With the development of genome sequencing and bioinformatics, new data have emerged. Prognosis-related gene signatures that were constructed using these new tools have made great contributions to tumor prognosis prediction. This study aimed to use bioinformatic methods to establish a prognostic metabolic-gene molecular model that can predict prognosis in patients with LUAD. This model could potentially guide personalized therapy for such patients.

## MATERIALS & METHODS

### Data expression datasets

The transcriptome expression profiles and corresponding clinical information for LUAD were downloaded from The Cancer Genome Atlas (TCGA; http://portal.gdc.cancer.gov/) and Gene Expression Omnibus (GEO; http://www.ncbi.nlm.nih.gov/geo/) databases. From the TCGA, gene expression data were of the HTSeq-FPKM type, obtained from

497 LUAD and 54 non-tumor samples. From the GEO, the GSE68465 dataset included 443 LUAD and 19 non-tumor samples, using the GPL96 platform (Affymetrix Human Genome U133A Array). The metabolic genes in the Kyoto Encyclopedia of Genes and Genomes (KEGG) pathway were extracted from Gene Set Enrichment Analysis (GSEA) (https://www.gsea-msigdb.org/gsea/index.jsp), and the overlapping metabolism-related genes were identified from TCGA and GSE68465 (*Possemato et al., 2011*; *Zhu et al., 2020*).

## Construction and validation of the prognostic metabolic gene signature

The clinical cases from the TCGA database were used to assess the prognostic associations of the metabolic genes with clinical outcomes. The differentially expressed genes (DEGs) between LUAD and paired non-tumor samples were obtained by the Wilcoxon rank sum test using the R package called "limma", and the adjusted *P*-value < 0.05 and absolute log2 fold change (FC) >1 were considered as the selection criterion. Univariate Cox regression analysis was used to identify prognosis-related metabolic genes, and adjusted *P*-values < 0.001 were considered statistically significant. The lasso penalty for Cox proportional hazards model (1,000 iterations) was used to construct the prognostic gene-expression signature utilizing an R package called "glmnet." The prognostic gene-expression signature was designed using a risk scoring method with the following formula:

$$Risk\ score = \sum_{i}^{n} (x_i * \beta_i)$$

where $x_i$ indicates the expression of gene $i$ and $\beta_i$ indicates the coefficient of gene $i$ generated from the Cox multivariate regression.

The R package "survminer" was used to explore the cutoff point of the risk score, which divided patients into high- and low-risk groups. The R package "survival" was used to draw the Kaplan–Meier survival curves to demonstrate the overall survival (OS) in the high- and low-risk groups. The R package "survival ROC" was used to evaluate the prognostic value of the gene-expression signature.

## Independence of the prognostic gene signature from other clinical characteristics

To determine whether the predictive power of the prognostic gene-expression signature could be independent from other clinicopathological variables in patients with LUAD (including age, sex, TNM stage, T stage, N stage, and M stage), univariate and multivariate Cox regression analyses were performed. The hazard ratio (HR), 95% confidence intervals (Cis), and *P*-values were calculated.

## Construction and validation of a predictive nomogram

The nomogram was constructed using all the independent prognostic factors of the Cox regression analyses using R package "rms." Validation of the nomogram was assessed by discrimination and calibration using the concordance index (C-index) by *Harrell, Lee & Mark (1996)* (bootstraps with 1,000 resamples) and the calibration plot, respectively.
### External validation of the prognostic metabolic gene signature

To verify the prognostic metabolic-gene molecular signature in the GEO dataset, the risk score of patients was calculated directly with the gene-expression signature constructed from the TCGA dataset for further analysis. The receiver operating characteristic (ROC) and Kaplan–Meier analyses were performed identically with the gene signature in the TCGA dataset. The mRNA expression levels of the signature genes were analyzed further using online databases (the Oncomine database (http://www.oncomine.org/) and TIMER database (http://cistrome.shinyapps.io/timer/)). The protein expression levels associated with the signature genes were validated using the Human Protein Atlas database (http://www.proteinatlas.org/). The known genetic alterations of the signature genes were investigated using cBioPortal for Cancer Genomics (http://www.cbioportal.org/).

### Gene set enrichment analysis

Enrichment analysis of the KEGG pathways of the signature genes was performed using GSEA on the TCGA dataset. The nominal (NOM) $P$-value < 0.05 and the False Discovery Rate (FDR) $q$-value <0.25 indicated statistical significance.

### Statistical analysis

All analyses were performed using R software v3.6.3 (R Foundation for Statistical Computing, Vienna, Austria). Two-tailed $P$-values < 0.05 were considered statistically significant.

## RESULTS

### Clinical characteristics

The TCGA dataset included 486 patients with LUAD (Table S1). The GEO dataset included 443 patients with LUAD (Table S1). Patients with a survival time of less than 30 days were omitted. For the study, 454 and 439 patients remained in the TCGA and GEO datasets, respectively. The detailed clinical characteristics of all patients are listed in Table 1.

### Building and validation of the prognostic metabolic gene signature

To clarify our study design, a flow chart of the analysis procedure is presented in Fig. 1. A list of 994 genes in the KEGG pathway was identified from GSEA (Table S2), and 633 overlapping metabolism-related genes were abstracted from TCGA and GSE68465 (Table S3). The 96 DEGs (72 up-regulated genes and 24 down-regulated genes) between LUAD and paired non-tumor samples were identified from the further analysis (Fig. 2; Table S4). Seven significant genes associated with OS were identified using univariate analysis (Table S4). Furthermore, six genes were selected to build the prognostic model using a lasso-penalized Cox analysis (Table 2). The six genes were phosphofructokinase platelet (*PFKP*), pyruvate kinase muscle (*PKM*), triosephosphate isomerase 1 (*TPI1*), lactate dehydrogenase A (*LDHA*), prostaglandin E synthase (*PTGES*), and thymidylate synthase (*TYMS*). Risk score = (0.00005× *PFKP* mRNA level) + (0.00173×*PKM* mRNA level) + (0.00038×*TPI1* mRNA level) + (0.00379×*LDHA* mRNA level) + (0.00292×*PTGES* mRNA level) + (0.02490× *TYMS* mRNA level).

**Table 1  Clinical characteristics of the included datasets.**

| Characteristics | TCGA (*n*, %) (*n* = 454) | GSE68465 (*n*, %) (*n* = 439) |
|---|---|---|
| Age | | |
| <60 | 133 (29.3%) | 128 (29.2%) |
| ≥60 | 321 (70.7%) | 311 (70.8%) |
| NA | 0 (0.0%) | 0 (0.0%) |
| Gender | | |
| Female | 248 (54.6%) | 218 (49.7%) |
| Male | 206 (45.4%) | 221 (50.3%) |
| NA | 0 (0.0%) | 0 (0.0%) |
| Grade | | |
| G1 | 0 (0.0%) | 60 (13.7%) |
| G2 | 0 (0.0%) | 206 (46.9%) |
| G3 | 0 (0.0%) | 166 (37.8%) |
| NA | 454 (100%) | 7 (1.6%) |
| TNM stage | | |
| I | 243 (53.5%) | |
| II | 105 (23.1%) | |
| III | 74 (16.3%) | |
| IV | 24 (5.3%) | |
| NA | 8 (1.8%) | 439 (100%) |
| T stage | | |
| T1 | 156 (34.4%) | 150 (34.2%) |
| T2 | 240 (52.9%) | 248 (56.5%) |
| T3 | 37 (8.1%) | 28 (6.4%) |
| T4 | 18 (4.0%) | 11 (2.5%) |
| Tx | 3 (0.7%) | 2 (0.5%) |
| N stage | | |
| N0 | 291 (64.1%) | 297 (67.7%) |
| N1 | 86 (18.9%) | 87 (19.8%) |
| N2 | 64 (14.1%) | 52 (11.8%) |
| N3 | 2 (0.4%) | 0 (0.0%) |
| Nx | 11 (2.4%) | 3 (0.7%) |
| M stage | | |
| M0 | 305 (67.2%) | 439 (100%) |
| M1 | 23 (5.1%) | 0 (0.0%) |
| Mx | 126 (27.8%) | 0 (0.0%) |
| Survival status | | |
| Alive | 300 (66.1%) | 206 (46.9%) |
| Dead | 154 (33.9%) | 233 (53.1%) |

**Notes.**
TCGA, The Cancer Genome Atlas.

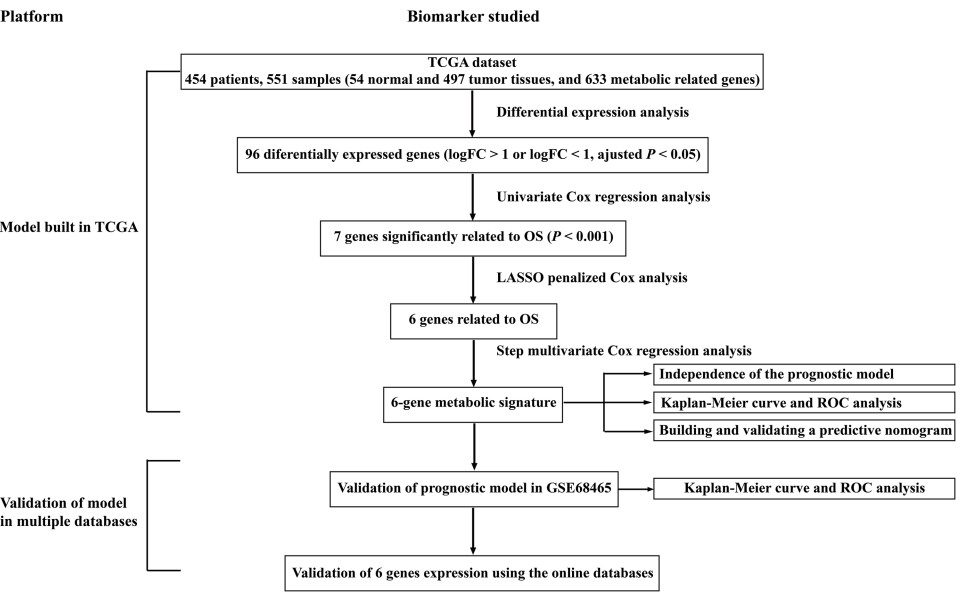

**Figure 1** **Overall flowchart of steps used in the construction of the prognostic metabolic gene signature.** The TCGA dataset was utilized to construct the prognostic metabolic gene signature. The TCGA clinical information, the GSE68465 dataset and online databases from international platforms were further utilized to validate the prognostic model. TCGA, The Cancer Genome Atlas; OS, overall survival; ROC, the receiver operating characteristic.

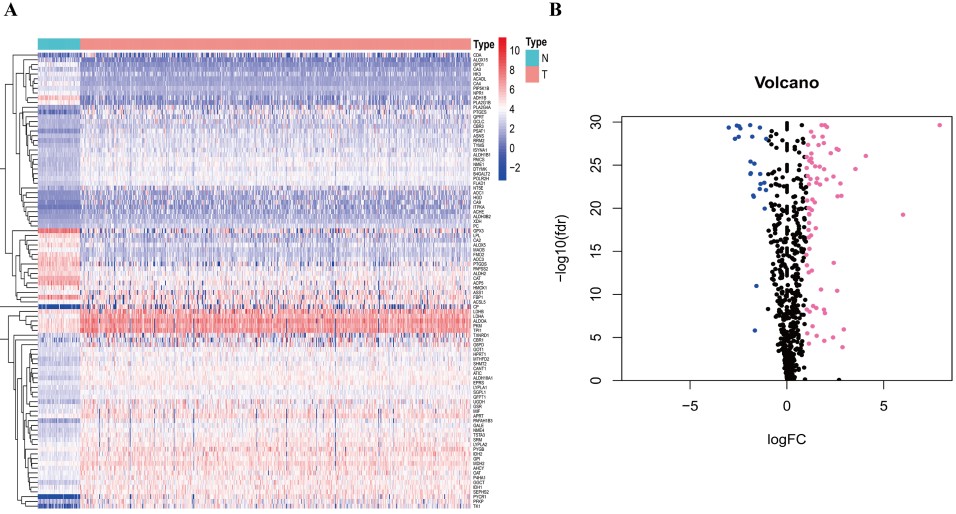

**Figure 2** **Heatmap and Volcano plot of metabolism-related DEGs.** (A) The heatmap of metabolism-related DEGs. The red color represented high expression genes, the blue color represented low expression genes, and the white color represented the expression genes with no significant difference (FDR < 0.05, absolute log FC > 1). (B) Volcano plot of metabolism-related DEGs. The pink, blue and black dots represented the high expression genes, low expression genes, and the expression genes with no significant difference (FDR < 0.05, absolute log FC > 1). DEGs, differentially expressed genes; FDR, false discovery rate.

**Table 2  Prognostic values for the six-gene metabolic signature in 454 LUAD patients.**

| Gene | Coef | HR | HR.95L | HR.95H | *P* value |
|------|------|-----|--------|--------|-----------|
| PFKP | 0.000050 | 1.009949 | 1.004184 | 1.015747 | 0.0007 |
| PKM | 0.001734 | 1.005149 | 1.002545 | 1.007759 | 0.000104 |
| TPI1 | 0.000384 | 1.003352 | 1.001479 | 1.005229 | 0.000448 |
| LDHA | 0.003792 | 1.005663 | 1.003774 | 1.007556 | 0.00000000396 |
| PTGES | 0.002922 | 1.008392 | 1.003408 | 1.0134 | 0.000946 |
| TYMS | 0.024904 | 1.033141 | 1.01572 | 1.050861 | 0.000172 |

**Notes.**
LUAD, lung adenocarcinoma; HR, hazard ratio; CI, confidence interval.

The 445 patients with LUAD were divided into the high-risk or low-risk group based on the median risk score of 0.861 in the TCGA dataset. Patients in the high-risk group had significantly poorer OS than those in the low-risk group ($P < 0.001$; Fig. 3A). The distribution of the risk score and survival status of the patients is presented in Fig. 3C, which showed a higher mortality in the high-risk group than in the low-risk group. The expression of the six prognostic genes is shown in the heatmap. All the six genes had a significant positive correlation with the high-risk group (Fig. 3E). The area under the curve (AUC) of the time-dependent ROC curve was used to identify the prognostic ability of the six-gene molecular signature. The AUCs of the six-gene signature model were 0.693, 0.655, and 0.565 for the 1-, 3-, and 5-year OS, respectively, suggesting that the prediction model had a good performance in predicting the OS in patients with LUAD (Fig. 3G).

The prognostic model was validated in the GSE68465 dataset. The 439 patients with LUAD were divided into the high-risk or low-risk group based on the median risk score of 0.861. Patients in the high-risk group had a poor OS compared with those in the low-risk group ($P < 0.001$; Fig. 3B). The distribution of the risk score and survival status showed a higher mortality in the high-risk group than in the low-risk group (Fig. 3D). The expression heatmap of the six prognostic genes showed that all the six genes had a significant positive correlation with the high-risk group (Fig. 3F). The AUCs of the six-gene signature model were 0.728, 0.654, and 0.618 for the 1-, 3-, and 5-year OS, respectively (Fig. 3H). Taken together, these results suggested that the prognostic model had a high sensitivity and specificity in predicting the OS in patients with LUAD.

## The prognostic gene signature was independent from other clinicopathological factors

Univariate and multivariate Cox regression analyses were conducted to assess the independent predictive value of the six-gene prognostic signature. In the TCGA dataset, univariate Cox regression analysis demonstrated that the prognostic model (HR: 2.845, *P* < 0.001), TNM stage (HR: 1.666, *P* < 0.001), T stage (HR: 1.605, *P* < 0.001), and N stage (HR: 1.806, *P* < 0.001) had a prognostic value for OS (Fig. 4A). Multivariate Cox regression analysis demonstrated that the only prognostic model (HR: 2.448, *P* <0.001) and TNM stage (HR: 1.950, *P* <0.01) were independent prognostic factors for OS (Fig. 4A). In the GSE68465 dataset, the prognostic model, T stage, N stage, and age had a prognostic value

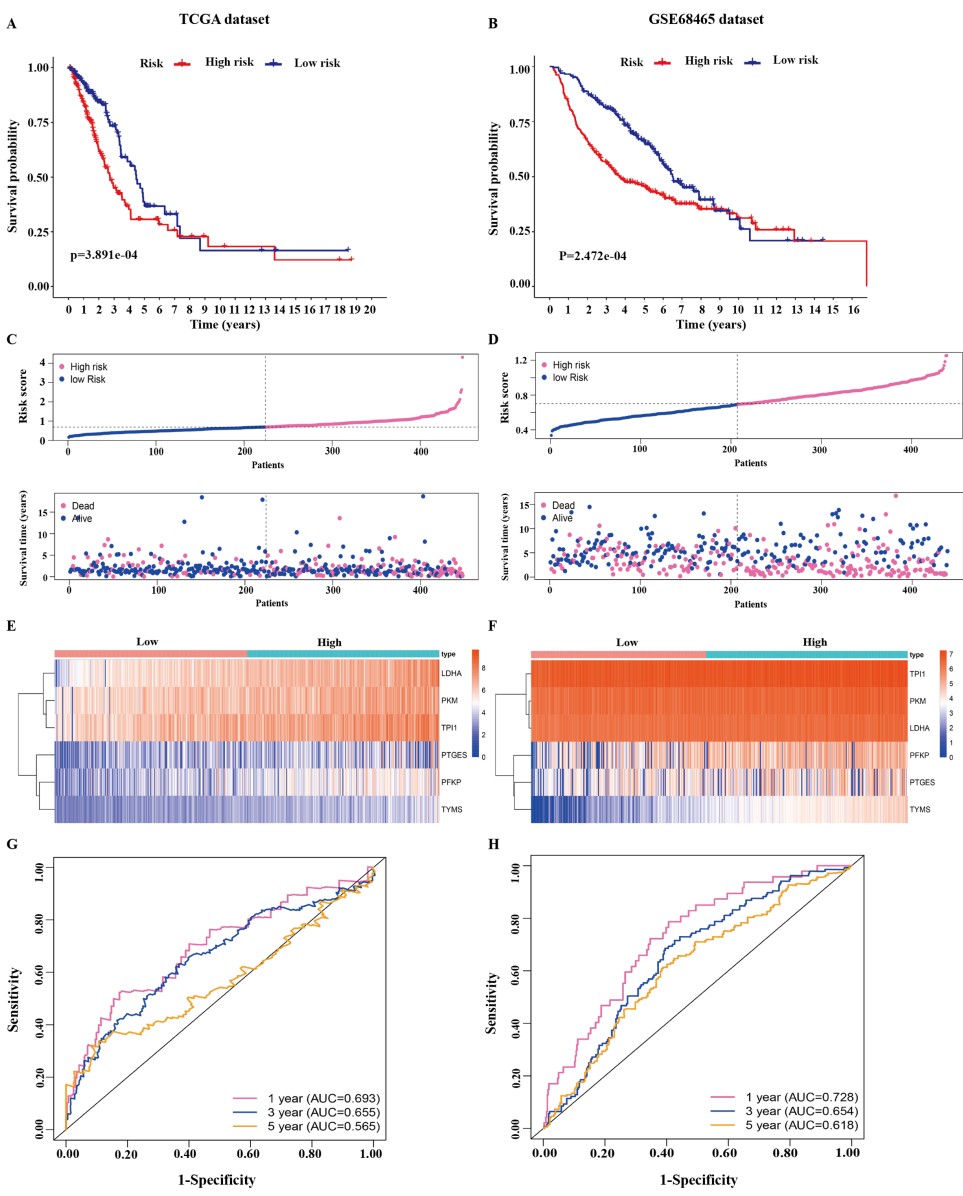

**Figure 3** **Identification of the prognostic model in lung adenocarcinoma.** (A, B) Kaplan–Meier curves of overall survival of the high-risk and low-risk groups stratified by the six-gene signature- based risk score in the TCGA or GEO dataset. (C, D) Risk score distribution, survival status distribution in the TCGA or GEO dataset. (E, F) The expression heatmap of the six prognostic genes in the TCGA or GEO dataset. (G, H) Time-dependent ROC curves of the six-gene signature in the TCGA or GEO dataset. TCGA, The Cancer Genome Atlas; GEO, Gene Expression Omnibus; ROC, receiver operating characteristic.

in the univariate and multivariate Cox regression analyses (Fig. 4B). Gender was the only independent prognostic factor for OS in the univariate Cox regression analysis (Fig. 4B).

In addition, the time-dependent ROC curve was used to identify the predictive ability of the prognostic model compared with the other clinicopathological characteristics. In the TCGA dataset, the AUCs of the prognostic model were 0.693, 0.655, and 0.565 for the 1-, 3-,

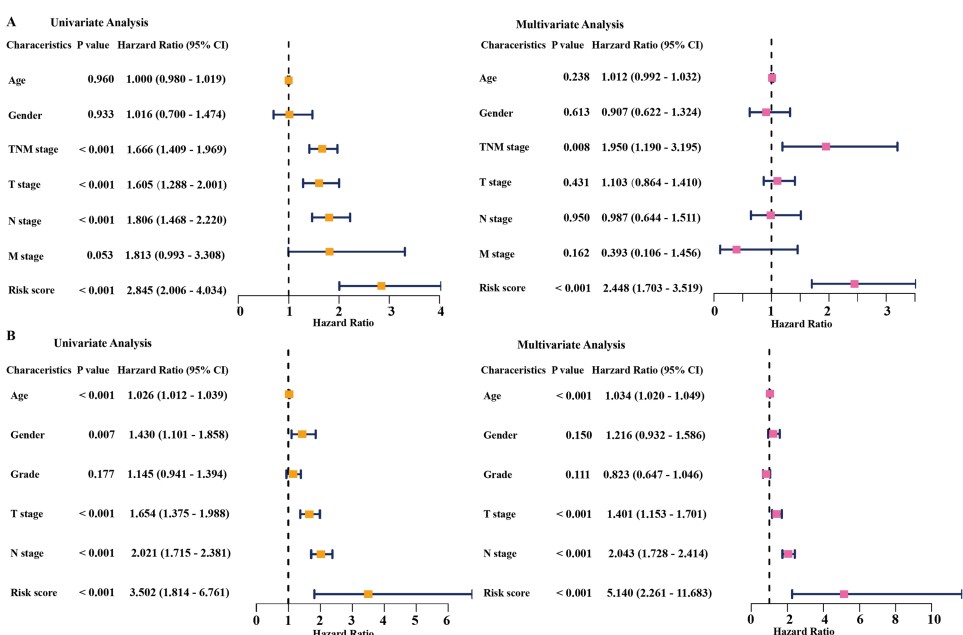

**Figure 4** Cox regression analysis of the associations between the prognostic model and clinicopathological characteristics with overall survival in LAUD. Univariate and multivariate Cox regression analyses in the TCGA dataset (A) and GEO dataset (B). LUAD, lung adenocarcinoma; TCGA, The Cancer Genome Atlas; GEO, Gene Expression Omnibus.

and 5-year OS, respectively, which were higher than most of the other clinicopathological characteristics including age (0.498, 0.511, 0.485), gender (0.579, 0.485, 0.451), T stage (0.673, 0.613, 0.608), N stage (0.685, 0.666, 0.628), and M stage (0.508, 0.527, 0.530) (Fig. 5A). Furthermore, in the GSE68465 dataset, the AUCs of the prognostic model were 0.728, 0.654, and 0.618 for the 1-, 3-, and 5-year OS, respectively, which were higher than most of the other clinicopathological characteristics including age (0.593, 0.568, 0.581), gender (0.539, 0.549, 0.547), grade (0.580, 0.571, 0.548), T stage (0.647, 0.606, 0.606), and N stage (0.690, 0.680, 0.655) (Fig. 5B). The prognostic model had a larger AUC value compared with other clinicopathological characteristics. These results indicated that the model was an excellent prognostic model for LAUD patients, especially for the 1- and 3-year OS.

These results suggested that our prognostic model could be an independent predictor of prognosis in patients with LAUD.

## Building and validating a predictive nomogram

A nomogram was built to predict the survival probability in patients with LAUD in the TCGA dataset. The nomogram was constructed using four prognostic factors (the TNM stage, T stage, N stage, and prognostic model; Fig. 6A). The C-index was calculated to evaluate the predictive ability of the nomogram for OS. The C-index for the nomogram was 0.754 (95% CI [0.561–0.947]). Calibration plots indicated that the nomogram had a good accuracy in predicting the 1- and 3-year OS (Fig. 6B).

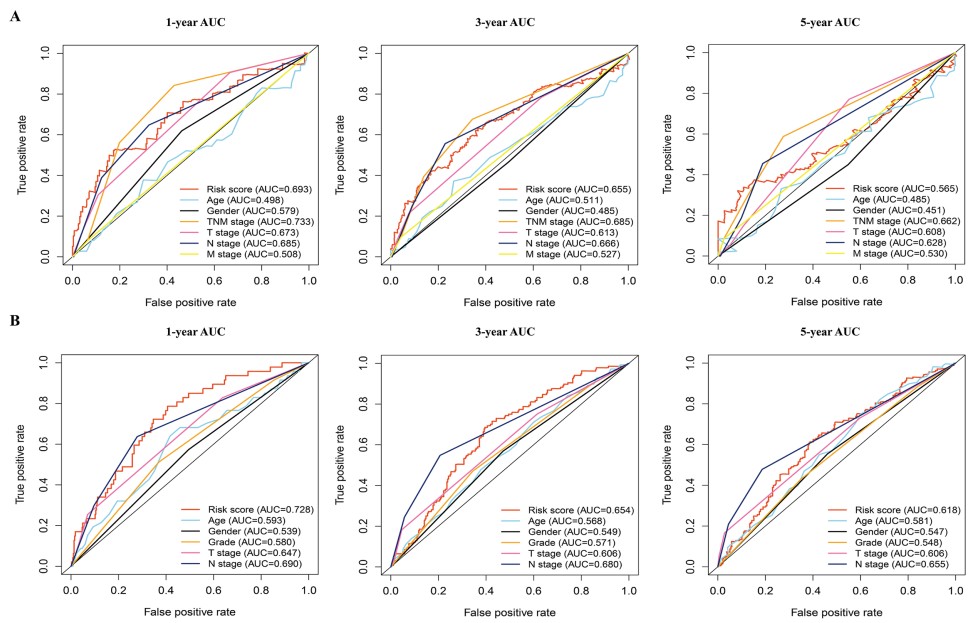

**Figure 5** **The time-dependent receiver operating characteristic (ROC) analysis for the prognostic model and clinicopathological characteristics in LAUD.** (A) The time-dependent ROC curves of risk score, age, gender, TNM stage, T stage, N stage, and M stage in the TCGA dataset. (B) The time-dependent ROC curves of risk score, age, gender, grade, T stage, and N stage in the GEO dataset. LUAD, lung adenocarcinoma.

To predict the survival probability more accurately, the combined prognostic model was built based on the nomogram. The combined prognostic model consisted of the TNM stage, T stage, N stage, and prognostic model. A time-dependent ROC curve was used to identify the predictive ability of the combined prognostic model. The AUCs of the combined prognostic models were 0.782, 0.717, and 0.688 for the 1-, 3-, and 5-year OS, respectively, which were higher than other clinical models including the TNM stage model (0.732, 0.687, 0.681), T stage model (0.671, 0.612, 0.613), N stage model (0.686, 0.661, 0.648), and the prognostic model (0.692, 0.634, 0.576). The combined model had the largest AUC value compared with other factors, which indicated that the combined model had a good predictive accuracy for survival. These results suggested that the predictive ability of the combined model built with the nomograms is better than other models, especially for predicting 1- and 3-year survival (Fig. 6C).

## Gene set enrichment analysis

To recognize signaling pathways that are differentially activated in LUAD, a GSEA was used, and a total of 49 significantly enriched KEGG pathways were found in the high-risk group and low-risk group (Table S5) of the TCGA dataset (FDR $q$-val < 0.25, NOM $p$-val < 0.05). Among them, many enriched pathways were related to metabolism and some highly dysregulated pathways including cell cycle, p53 signaling pathway, and basal transcription factors were also contained in these results (Table S5). We chose the top five significantly enriched metabolism-signaling pathways depending on the normalized

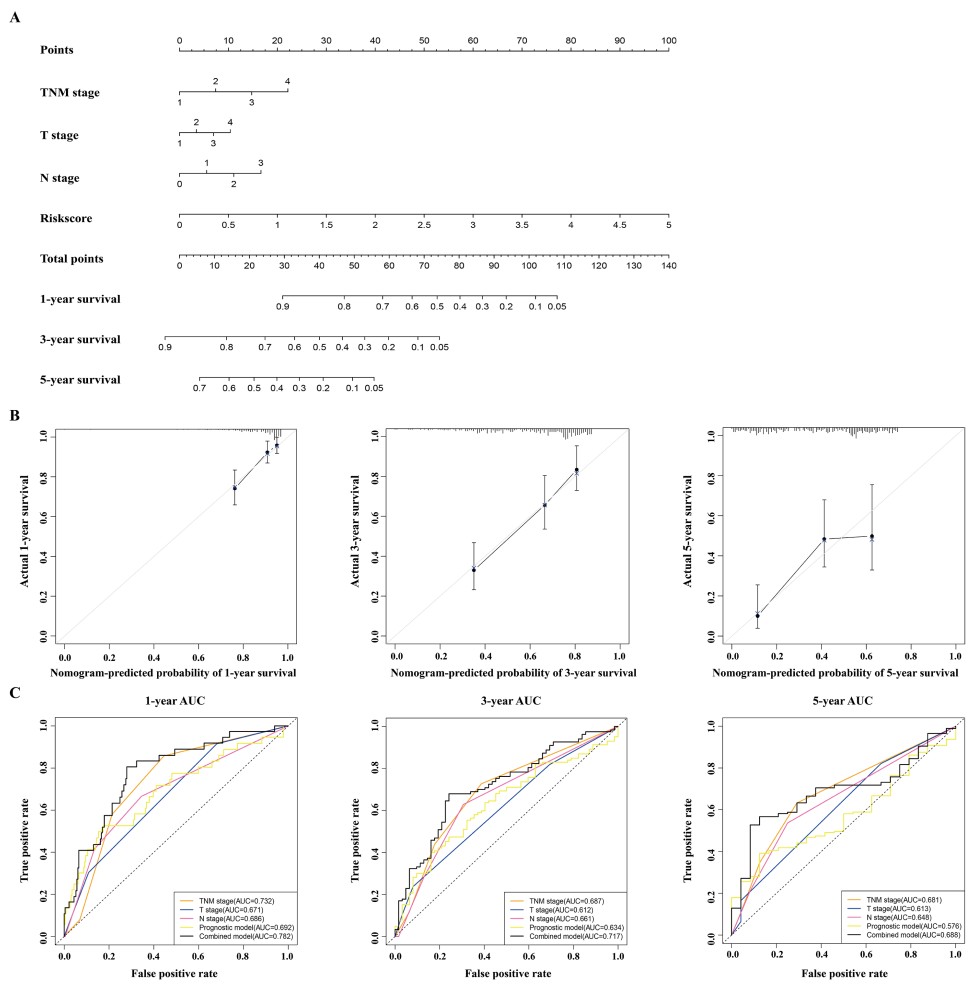

**Figure 6** **Construction and validation of a nomogram for survival prediction in LUAD from the TCGA dataset.** (A) The nomogram was built in the TCGA dataset. (B) Calibration plots revealed the nomogram-predicted survival probabilities. (C) The time-dependent ROC analysis evaluated the accuracy of the nomogram. TCGA, The Cancer Genome Atlas; ROC, receiver operating characteristic; LUAD, lung adenocarcinoma.

enrichment score from the high-risk group or low-risk group. We found that the top five most significantly enriched metabolism-related pathways of the high-risk group were the cysteine and methionine, fructose and mannose, glyoxylate and dicarboxylate, purine, and pyrimidine pathways (Fig. 7A). The top five most significantly enriched metabolism-related pathways of the low-risk group were the alpha linolenic acid, arachidonic acid, ether lipid, glycerophospholipid, and linoleic acid pathways (Fig. 7B). Most of the metabolism-related pathways in the high-risk group mainly focused on amino acid and glycolysis metabolism, while the pathways in the low-risk group mainly focused on lipid metabolism. The results of the ten representative enriched metabolism-related KEGG pathways are given in Table 3. Furthermore, all the six metabolic genes of the prognostic model enriched these metabolism pathways significantly. *LDHA* enriched the cysteine and methionine pathway (Table S6);

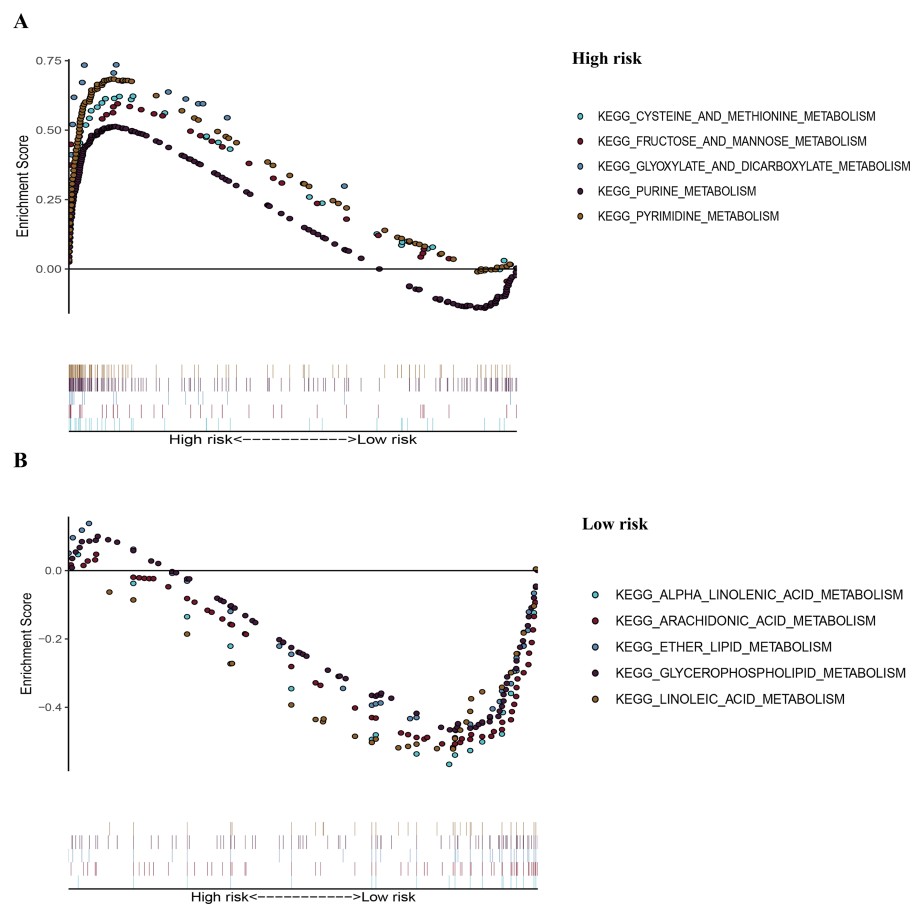

**Figure 7   The representative enriched metabolism-related KEGG pathways in the TCGA dataset by GSEA.** (A) The top five significantly representative enriched metabolism-related KEGG pathways in the high-risk group. (B) The top five significantly representative enriched metabolism-related KEGG pathways in the low-risk group. Related parameters for the ten representative enriched metabolism-related KEGG pathways are given in Table 3. GSEA, Gene Set Enrichment Analysis; KEGG, Kyoto Encyclopedia of Genes and Genomes; TCGA, The Cancer Genome Atlas.

*PFKP* and *TPI1* enriched the fructose and mannose pathway (Table S6); *PKM* enriched the purine pathway (Table S6); *TYMS* enriched the pyrimidine pathway (Table S6); and *PTGES* enriched the arachidonic acid pathway (Table S6). The results further elucidated the role of metabolism in LUAD and the value of the six-gene signature in predicting the prognosis of LUAD.

## External validation using online databases

To further identify the role of the six metabolic genes in LUAD, we compared the mRNA expression levels of the six metabolic genes (*PFKP*, *PKM*, *TPI1*, *LDHA*, *PTGES*, and *TYMS*) in the LAUD tissues with those in the normal lung tissues using data from the Oncomine database (Fig. 8). Obviously, all the six genes were overexpressed in lung cancer in all the datasets from the Oncomine database with the threshold of fold change = 2, *P*-value = 0.001 (Fig. 8A). Furthermore, the mRNA levels of all the six genes in LUAD were

**Table 3  The results of the ten representative enriched metabolism-related KEGG pathways analysed by GSEA.**

| Pathway | Size | ES | NES | NOM p-val | FDR q-val |
|---|---|---|---|---|---|
| High risk | | | | | |
| KEGG_CYSTEINE_AND_METHIONINE_ METABOLISM | 34 | 0.62 | 1.98 | 0.00 | 0.006 |
| KEGG_FRUCTOSE_AND_MANNOSE_ METABOLISM | 33 | 0.60 | 1.95 | 0.002 | 0.007 |
| KEGG_GLYOXYLATE_AND_DICARBOXYLATE_ METABOLISM | 16 | 0.74 | 1.89 | 0.002 | 0.013 |
| KEGG_PURINE_METABOLISM | 157 | 0.51 | 2.02 | 0.000 | 0.005 |
| KEGG_PYRIMIDINE_METABOLOSM | 98 | 0.68 | 2.36 | 0.000 | 0.000 |
| Low risk | | | | | |
| KEGG_ALPHA_LINOLENIC_ACID_METABOLISM | 19 | −0.60 | −1.82 | 0.002 | 0.113 |
| KEGG_ARACHIDONIC_ACID_METABOLISM | 58 | −0.53 | −1.86 | 0.000 | 0.101 |
| KEGG_ETHER_LIPID_ METABOLISM | 33 | −0.51 | −1.73 | 0.011 | 0.129 |
| KEGG_GLYCEROPHOSPHOLIPID_METABOLISM | 77 | −0.48 | −1.91 | 0.002 | 0.127 |
| KEGG_LINOLEIC_ACID_ METABOLISM | 29 | −0.55 | −1.77 | 0.008 | 0.138 |

**Notes.**

KEGG, Kyoto Encyclopedia of Genes and Genomes; GSEA, Gene Set Enrichment Analysis; ES, enrichment score; NOM $p$-val, nominal $p$-value; FDR $q$-val, false discovery rate $q$-value; NES, normalized enrichment score.

significantly upregulated than those in normal tissues in the combined LUAD datasets from the Oncomine database (Fig. 8B; Table 4). To further validate the overexpression of the six genes in LUAD, we analyzed the expression of the six genes using TIMER databases (Fig. 9). The results revealed that all the mRNA expression of the six genes in LUAD were significantly higher than in normal tissues. All the results from the Oncomine and TIMER databases were consistent with our results for the TCGA and GEO datasets. In addition, the mRNA expression of the six genes was also higher in esophageal carcinoma, head and neck squamous cell carcinoma, lung squamous cell carcinoma, and stomach adenocarcinoma from the TIMER databases (Fig. 9). The protein expressions of these six genes were analyzed using clinical specimens from the Human Protein Profiles (Figs. 10A and 10B; Table 5). The representative images of the six gene protein levels from the Human Protein Profiles are shown in Fig. 10A. Compared with the expression level in normal lung tissue, *LDHA* (100%, $n = 7$) and *TYMS* (80%, $n = 5$) showed a significantly higher percentage of high/medium expression levels in the LAUD tissue (Fig. 10B; Table 5). *PKM* (50%, $n = 6$), *PFKP* (33.33%, $n = 6$), and *PTGES* (16.67%, $n = 6$) showed a significant moderate percentage of high/medium expression levels in the LAUD tissue (Fig. 10B; Table 5). However, *TPI1* showed no detected expression both in the LAUD and normal lung tissue (Fig. 10B; Table 5). The genetic alterations were explored in the cBioPortal database. Amplifications and mutations were the most common alterations in the six metabolic genes (Fig. 10C). The aberrant genetic alterations might elucidate the overexpression of these six genes in LUAD.

Altogether, the correlation of the aberrant expression of these six genes with LAUD cancer was further validated using multiple online databases.

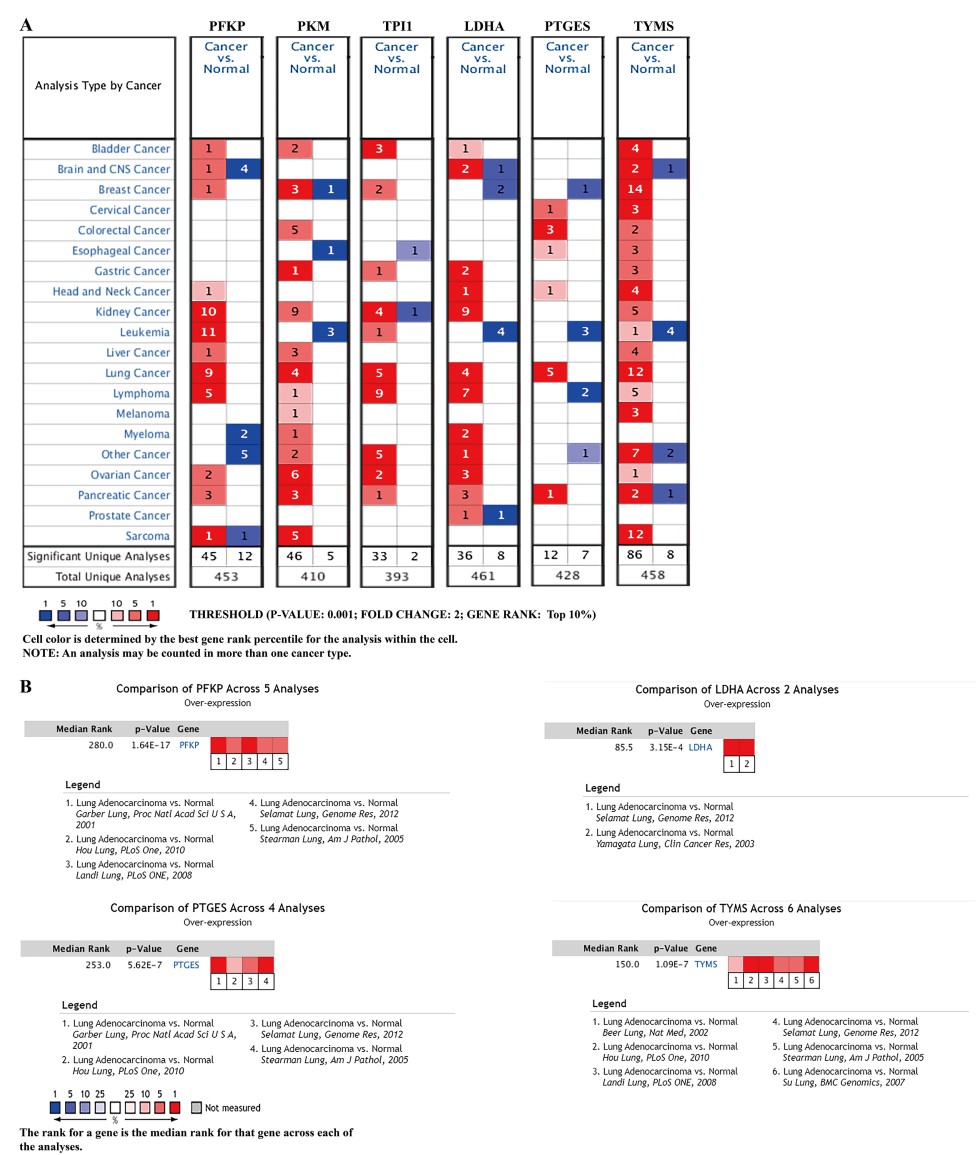

**Figure 8  mRNA expression levels of the six prognostic genes from online databases.** (A) mRNA expression levels of the six genes in the Oncomine database (http://www.oncomine.org/). The threshold is shown at the bottom (*P* value < 0.001 and fold change > 2 were utilized for screening). The figure in the colored cell represents the number of datasets complying with the threshold. The red cells indicate that the genes were overexpressed in the cancer, while the blue cells indicate that the genes were overexpressed in the normal tissues. (B) Comparisons of the mRNA expression levels of the six genes between LUAD and normal tissues in the combined LUAD datasets from the Oncomine database. *PFKP*, phosphofructokinase platelet; *PKM*, pyruvate kinase muscle; *TPI1*, triosephosphate isomerase 1; *LDHA*, lactate dehydrogenase A; *PTGES*, prostaglandin E synthase; *TYMS*, thymidylate synthase; LUAD, lung adenocarcinoma.

**Table 4  Comparison of mRNA expression levels of the six genes between LUAD and normal tissues from the Oncomine database.**

| Gene | Analysis type of lung cancer vs. normal | t-Test | Fold change | P value | References |
|------|------------------------------------------|--------|-------------|---------|------------|
| PFKP | LUAD ($n = 40$) vs. Normal ($n = 6$) | 7.146 | 3.469 | 7.56E−8 | *Garber et al. (2001)* |
| | LUAD ($n = 58$) vs. Normal ($n = 49$) | 11.177 | 2.685 | 4.35E−19 | *Landi et al. (2008)* |
| | LUAD ($n = 45$) vs. Normal ($n = 65$) | 7.946 | 2.536 | 3.39E−11 | *Hou et al. (2010)* |
| | LUAD ($n = 58$) vs. Normal ($n = 58$) | 10.910 | 2.883 | 1.64E−17 | *Selamat et al. (2012)* |
| | LUAD ($n = 20$) vs. Normal ($n = 19$) | 5.277 | 3.149 | 9.37E−6 | *Stearman et al. (2005)* |
| | Comparison of PFKP expression across 5 Analysis between LUAD and Normal | – | – | 1.64E−17 | – |
| PKM | LUAD ($n = 58$) vs. Normal ($n = 58$) | 12.037 | 2.551 | 3.56E−20 | *Selamat et al. (2012)* |
| TPI1 | LUAD ($n = 40$) vs. Normal ($n = 6$) | 4.929 | 2.283 | 4.03E−4 | *Garber et al. (2001)* |
| LDHA | LUAD ($n = 9$) vs. Normal ($n = 3$) | 4.502 | 4.037 | 6.29E−4 | *Yamagata et al. (2003)* |
| | LUAD ($n = 58$) vs. Normal ($n = 58$) | 11.533 | 2.179 | 1.59E−19 | *Selamat et al. (2012)* |
| | Comparison of LDHA expression across 2 Analysis between LUAD and Normal | – | – | 3.15E−4 | – |
| PTGES | LUAD ($n = 20$) vs. Normal ($n = 19$) | 9.332 | 5.883 | 1.54E−11 | *Stearman et al. (2005)* |
| | LUAD ($n = 40$) vs. Normal ($n = 6$) | 6.690 | 4.969 | 1.12E−6 | *Garber et al. (2001)* |
| | LUAD ($n = 58$) vs. Normal ($n = 58$) | 10.267 | 2.179 | 5.58E−16 | *Selamat et al. (2012)* |
| | LUAD ($n = 45$) vs. Normal ($n = 65$) | 6.513 | 2.170 | 6.22E−9 | *Hou et al. (2010)* |
| | Comparison of PTGES expression across 4 Analysis between LUAD and Normal | – | – | 5.62E−7 | – |
| TYMS | LUAD ($n = 45$) vs. Normal ($n = 65$) | 9.322 | 3.929 | 6.92E−15 | *Hou et al. (2010)* |
| | LUAD ($n = 27$) vs. Normal ($n = 30$) | 7.395 | 3.016 | 2.40E−9 | *Su et al. (2007)* |
| | LUAD ($n = 58$) vs. Normal ($n = 49$) | 11.169 | 2.797 | 9.86E−20 | *Landi et al. (2008)* |
| | LUAD ($n = 20$) vs. Normal ($n = 19$) | 6.509 | 2.118 | 2.18E−7 | *Stearman et al. (2005)* |
| | LUAD ($n = 86$) vs. Normal ($n = 10$) | 4.191 | 2.158 | 3.05E−4 | *Beer et al. (2002)* |
| | LUAD ($n = 58$) vs. Normal ($n = 58$) | 8.565 | 2.040 | 3.35E−13 | *Selamat et al. (2012)* |
| | Comparison of TYMS expression across 6 Analysis between LUAD and Normal | – | – | 1.09E−7 | – |

**Notes.**
Owing to only one dataset meeting the screening criteria, the comparison of PKM or TPI1 expression in LUAD and normal has not been built based on the combined LUAD datasets. *P* value < 0.001 and fold change > 2 were utilized for screening.
LUAD, lung adenocarcinoma.

## DISCUSSION

LUAD is the most common histological subtype of primary lung cancer. The incidence of LUAD has been increasing rapidly, and mortality has not significantly decreased despite great improvements in research and treatment. Therefore, exploring the molecular mechanisms of LUAD progression and constructing a valid and accurate molecule-based tool for evaluating the prognosis in patients is urgently needed. This could help design more efficient therapeutic strategies for LUAD. Metabolic reprogramming in cancers could lead to their development and progression (*Nwosu et al., 2017*; *Liu et al., 2020*). Characterization of the changes in metabolic gene expression in LUAD would allow development of novel prognostic biomarkers. However, a single biomarker is not a robust measure for predicting patient prognosis. Thus, constructing a robust multiple-biomarker signature for predicting the prognosis in cancer patients is necessary.

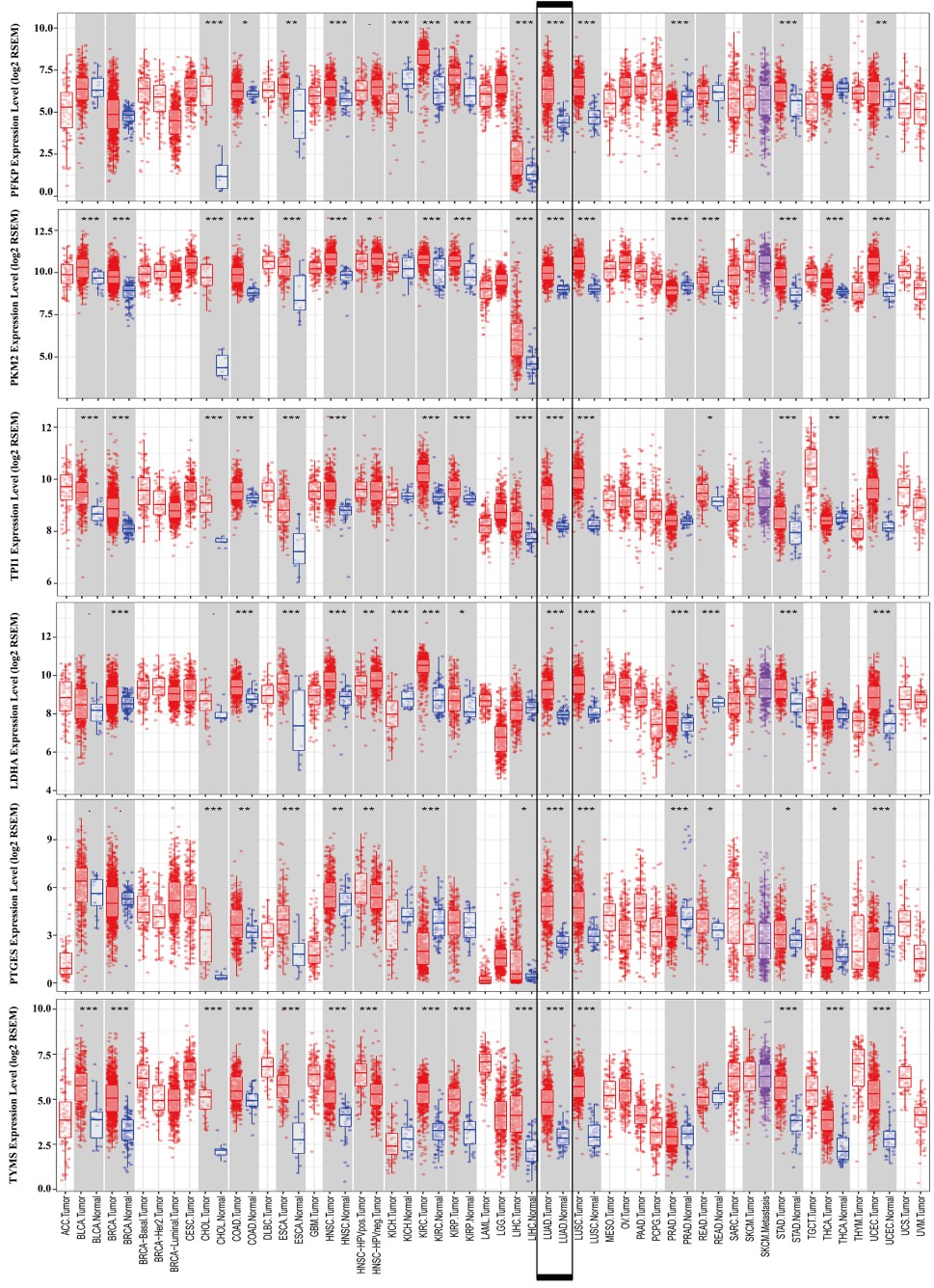

**Figure 9** **mRNA expression levels of the six prognostic genes extracted from online database.**
The mRNA expression levels of the six genes in different tumour types from the TIMER database
(http://cistrome.shinyapps.io/timer/) (*P < 0.05, **P < 0.01, ***P < 0.001). *PFKP*, phosphofructokinase
platelet; *PKM*, pyruvate kinase muscle; *TPI1*, triosephosphate isomerase 1; *LDHA*, lactate dehydrogenase
A; *PTGES*, prostaglandin E synthase; *TYMS*, thymidylate synthase.

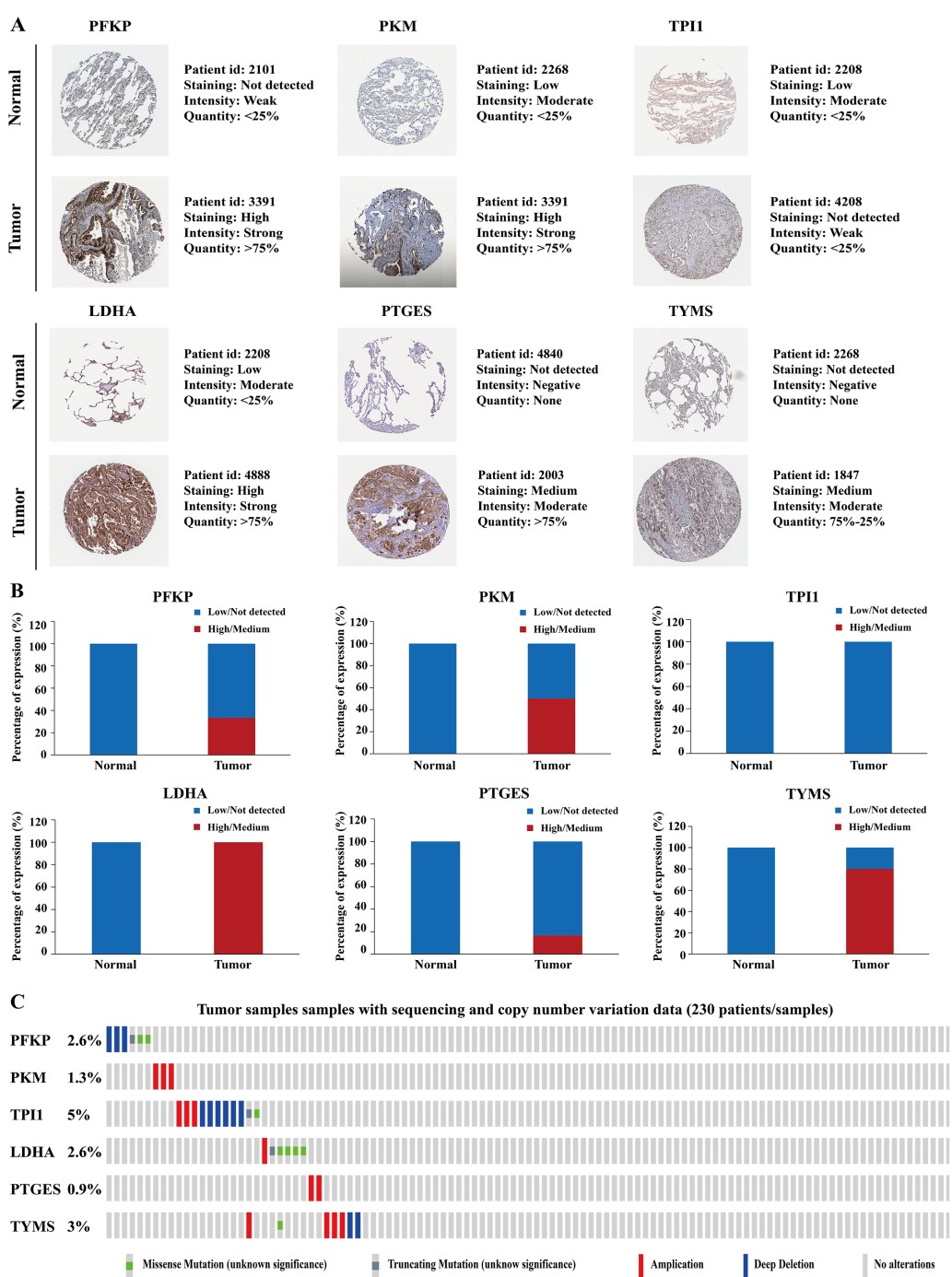

**Figure 10 Protein expression levels and genetic alterations of the corresponding six prognostic genes obtained from online databases.** (A) The representative immunohistochemistry images of the protein expression of the six genes in the normal lung tissues and LUAD tissues from the Human Protein Atlas database (http://www.proteinatlas.org/). (B) The percentage of protein expression levels in the normal lung tissues and LUAD tissues analysed based on the Human Protein Atlas database. Anti-PFKP antibody is HPA018257; 

**Figure 10 (…continued)**
anti-PKM antibody is CAB019421; anti-TPI1 antibody is HPA053568; anti-LDHA antibody is CAB069404; anti-PTGES is HPA045064; anti-TYMS antibody is CAB002784. (C) Genetic alterations of the six genes in 230 LUAD patients / samples (TCGA, Firehose Legacy). Data were obtained from the cBioportal for Cancer Genomics (http://www.cbioportal.org/). *PFKP*, phosphofructokinase platelet; *PKM*, pyruvate kinase muscle; *TPI1*, triosephosphate isomerase 1; *LDHA*, lactate dehydrogenase A; *PTGES*, prostaglandin E synthase; *TYMS*, thymidylate synthase; TCGA, The Cancer Genome Atlas; LUAD, lung adenocarcinoma.

**Table 5** Protein expression levels of the six prognostic genes in the normal lung tissues and LUAD tissues obtained from the Human Protein Atlas database.

| Gene name | Tissue type | Patients in high/medium staining $n$ (%) | Patients in low/not detected staining $n$ (%) |
|---|---|---|---|
| PFKP | Normal | 0 (0%) | 3 (100%) |
|  | Tumor | 2 (33.33%) | 4 (66.67%) |
| PKM | Normal | 0 (0%) | 3 (100%) |
|  | Tumor | 3 (50%) | 3 (50%) |
| TPI1 | Normal | 0 (0%) | 3 (100%) |
|  | Tumor | 0 (0%) | 3 (100%) |
| LDHA | Normal | 0 (0%) | 3 (100%) |
|  | Tumor | 7 (100%) | 0 (0%) |
| PTGES | Normal | 0 (0%) | 3 (100%) |
|  | Tumor | 1 (16.67%) | 5 (83.33%) |
| TYMS | Normal | 0 (0%) | 3 (100%) |
|  | Tumor | 4 (80%) | 1 (20%) |

**Notes.**
LUAD, lung adenocarcinoma.

We identified and designed a novel six-gene prognostic molecular signature based on the TCGA database and validated its efficiency in the GSE68465 dataset. The results indicated that the molecular signature was significantly associated with OS in patients with LUAD in the training and validation sets. These results indicate that the molecular signature has a robust prognostic value, especially for predicting short-term survival in patients with LUAD. These results also demonstrated that the prognostic signature was independent of other clinicopathological characteristics, which further supports the prognostic value of this signature.

To increase the accuracy of the prediction of prognosis, we constructed a nomogram built with the combination of genetic and clinically related variables of patients with LUAD. The nomogram included the prognostic model, TNM, T stage, and N stage. Its predictive accuracy was verified using calibration plots, the C-index, and the AUC, which indicated that the nomogram had a greater predictive value than the previous systems. The Gene Set Enrichment Analysis showed that many significantly enriched pathways were metabolism-related pathways. The different risk groups possessed different metabolic pathway features. The metabolism-related pathways in the high-risk group were mainly associated with amino acid and glycolysis metabolism, while the pathways in the low-risk group were mainly associated with lipid metabolism. These results revealed that the

different risk groups possessed the different metabolic features, which might provide the underlying metabolic mechanisms of promoting the prognosis of LUAD. All these results further suggest a strong association between the molecular signature and metabolic systems and might reflect the dysregulated metabolic microenvironment of cancers.

Most of the six genes in our prognostic signature are suggested to be related to cancer development. *PFKP* is a major isoform of cancer-specific phosphofructokinase-1, an enzyme that catalyzes the phosphorylation of fructose-6-phosphate to form fructose-1,6-bisphosphate. Recently, *PFKP* was noted to have an aberrant upregulation in many cancers, such as breast cancer, prostate cancer, and glioblastoma. The dynamic upregulation of *PFKP* promotes metabolic reprogramming and cancer cell survival (*Bjerre et at., 2019*; *Kim et al., 2017*). As a key regulator enzyme in glycolysis, *PFKP* enriched the fructose and mannose metabolism pathway. Recent studies showed that *PFKP* is highly expressed in lung cancer and promotes lung cancer development via fructose and mannose metabolism (*Shen et al., 2020*; *Wang et al., 2015*). *PKM* is a rate-limiting enzyme in the final step of glycolysis, that is considered as one of the metabolic hallmarks of cancer (*Prakasam et al., 2017*). The abnormal expression of *PKM* promoted cancer growth, invasion, and metastasis by governing aerobic glycolysis (*Prakasam et al., 2017*; *Zahra et al., 2020*) and induced cancer treatment resistance (*Calabretta et al., 2016*). Furthermore, *PKM* is overexpressed in non-small cell lung cancer (NSCLC) and involved in the development and prognosis of NSCLC (*Luo et al., 2018*). *TPI1* is a crucial enzyme in carbohydrate metabolism, catalyzing the interconversion of dihydroxyacetone phosphate and d-glyceraldehyde-3-phosphate during glycolysis and gluconeogenesis. *TPI1* is abnormally expressed in different kinds of cancers, such as breast cancer, gastric cancer, and lymphoma and is associated with a poor prognosis in patients with neuroblastoma and pancreatic cancer through dysregulating glycometabolism (*Ludvigsen et al., 2018*; *Applebaum et al., 2016*; *Follia et al., 2019*). *LDHA* is an enzyme that catalyzes the interconversion of pyruvate and lactate. *LDHA* was enriched in cysteine and methionine metabolism, and its aberrant metabolism regulation promoted many pathological processes in tumors, such as cell proliferation, survival, invasion, metastasis, and immunity (*Dorneburg et al., 2018*). Overexpressed *LDHA* is associated with poor prognosis in many tumors, including NSCLC, breast cancer, gallbladder carcinoma, and gastrointestinal cancer (*Mizuno et al., 2020*; *Guddeti et al., 2019*). *PTGES* is a key enzyme in the arachidonic acid metabolism pathway. An abnormally high expression of *PTGES* is correlated with proliferation, invasion, and metastasis in many cancer cells (*Kim et al., 2016*; *Delgado-Goñi et al., 2020*). The dysregulated *PTGES* promoted tumor migration and metastasis of lung cancer cells and played an important role in lung cancer progression (*Wang et al., 2019*). *TYMS is* a rate-limiting enzyme, which plays an important role in regulating the pyrimidine metabolism signaling pathway (*Yeh et al., 2017*). *TYMS* is overexpressed frequently in different kinds of cancers, such as NSCLC, pancreatic, colorectal, and breast cancers, and it has resulted in a poor cancer prognosis and chemotherapy resistance via dysregulating pyrimidine metabolism (*TroncarelliFlores et al., 2019*; *Wu et al., 2019*). In our study, we constructed a six-gene signature for a prognostic model based on the TCGA database. This novel six-gene signature had a higher survival prediction, and the predictive ability of this signature was further validated by the

GSE68465 dataset and multiple online databases. To our knowledge, the six-gene signature for prognosis prediction in LUAD has not been reported yet. Compared with the traditional prognostic models such as clinical characteristics (e.g., TNM stage, vascular tumor invasion, and organization classification) or a single molecular biomarker, a multi-gene signature can predict the prognosis more accurately and provide a clearer molecular mechanism for personalized LUAD therapy.

There are limitations in our study. First, our nomogram was not validated further in the GEO database because the GSE68465 lacked detailed TNM stage data. Thus, the nomogram should be externally validated using larger datasets from multicenter clinical trials and perspective studies. Second, functional experiments should be further performed to explore the molecular mechanisms predicted by the metabolic gene expression.

# CONCLUSIONS

We concluded from our research results that the six-gene metabolic prognostic signature could accurately predict the prognosis in patients with LUAD. The molecular signature may provide potential biomarkers for metabolic therapy and prognosis prediction of LUAD.

## Funding

This work supported by the Science and Technology Project of Liaoning Province (No. 2019-ZD-0753 and No.2018010137-301). The funders had no role in study design, data collection and analysis, decision to publish, or preparation of the manuscript.

## Grant Disclosures

The following grant information was disclosed by the authors:
Science and Technology Project of Liaoning Province: 2019-ZD-0753, 2018010137-301.

## Competing Interests

The authors declare there are no competing interests.

## Author Contributions

- Yubo Cao conceived and designed the experiments, performed the experiments, analyzed the data, prepared figures and/or tables, authored or reviewed drafts of the paper, and approved the final draft.
- Xiaomei Lu performed the experiments, analyzed the data, authored or reviewed drafts of the paper, and approved the final draft.
- Yue Li and Xiulin Li performed the experiments, analyzed the data, prepared figures and/or tables, and approved the final draft.
- Jia Fu, Hongyuan Li and Ziyou Chang analyzed the data, prepared figures and/or tables, and approved the final draft.
- Sa Liu conceived and designed the experiments, authored or reviewed drafts of the paper, and approved the final draft.

## Data Availability

The raw measurements are available in the Supplemental Files.

## Supplemental Information

Supplemental information for this article can be found online at http://dx.doi.org/10.7717/peerj.10320#supplemental-information.

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
