# Peer review of "Identification of a six-gene metabolic signature predicting overall survival for patients with lung adenocarcinoma"

_PeerJ, doi:10.7717/peerj.10320_

## Round 0.1 · original submission · Minor Revisions

You have used a few databases and bioinformatics tools to identify a six-gene metabolic signature predicting overall survival for patients with lung adenocarcinoma. The paper is well written and present interesting findings. However, I agree with both reviewers, the presentation of some of the results needs to be improved before the paper is published. You will need to address the recommendations of the reviewers and the list below before the paper is accepted for publication:

1) State the names of the genes in the abstract.
2) Fully describe your figures/graphs in the results section.
3) The table showing the six genes and their expression values in normal and cancer samples should be included in the paper and not as supplementary table.
4) Figure 1 needs legend
5) Figure 3 legend is not satisfactory. You should give provide an explanatory legend and explain what each of this graphs/figure show. Panel A and B consist of five graphs each, and it is not clear what each of these figures is showing.
6) Figure 7 provide more explanation in the legend and briefly describe what you are showing.
7) Figure 8: legend needs more details for both panels. Oncomine results should be presented in more accessible format not just screenshots. Can these data be extracted and presented in graphs?
8) Figure 9 legend needs more detail. You should try to extract the relevant data from this figure or try to compare the expression of these six genes to other cancers. Is the increase expression of these six genes specific to lung adenocarcinoma?
9) Figure 10: provide more details to the legend. Extract data and show in form of graph.
10) Provide more details about the pathways and the role of these genes in these pathways.

Reviewer 1 ·

Basic reporting

No comment

Experimental design

No comment

Validity of the findings

The figures are a bit crowded and would benefits of a bit more explanation and additional legend descriptions.

Some additional files could be merge as some just contains 3 or 4 entries on a excel sheet.

Additional comments

The paper is well written and provide analysing supporting the conclusions.
This work tries fo fill a gap in the study of LUAD prognosis and thus propose to face a valable research question.
The authors made a good use of available databases and performed relevant analysis in order to identify these genes.
More details about the metabolic pathways involved and how these 6 genes could interfere would bring more strength to this manuscript.

Reviewer 2 ·

Basic reporting

No comment.

Experimental design

No comment.

Validity of the findings

No comment.

Additional comments

In the manuscript ‘Identification of a six-gene metabolic signature predicting overall survival for patients with lung adenocarcinoma’, using a bioinformatics approach the authors provide some interesting insights concerning the role of a six-gene metabolic signature in the prediction of survival in LUAD patients. Overall, it is a robustly conducted study and a well-written manuscript. However, some minor corrections need to be made before it is accepted to be published.

1. The world ‘prognoses’ should be substituted by ‘prognosis’ throughout the text.
2. In the ‘Gene Set Enrichment Analysis’ section of the results, the metabolism-related enriched pathways should be described further, including some of the key genes involved.
3. Tables S6 and S7 should be moved in the main text, potentially as one merged table.
4. Figure 1 should have at least a brief legend.
5. The legend in figure 3 should provide some more details, i.e. what each graph corresponds to in each panel (A and B). A clearer alignment of the graphs would be useful, as well.
6. The quality of figure 8 should be improved. The right panel is almost impossible to read without zooming in. Alternatively, some of the data could be presented in the form of a table.
7. Figure 9 is redundantly complicated. It would be clearer if the figure was focused only on LUAD, instead of incorporating so many different cancer types. It may be worth keeping only some relevant cancer types, e.g. other lung cancers or cancers where LUAD could metastasise to.
8. The legend of figure 10 needs elaboration/details. Show which protein shows elevated/ decreased expression for each corresponding transcript of the six-gene signature.

---

## Round 0.2 · accepted · Accept

I am pleased to let you know that, based on your revisions, your manuscript has now been accepted as an article of PeerJ.